# Disturbance Observer-Based Sliding Mode Controller for Underwater Electro-Hydrostatic Actuator Affected by Seawater Pressure

Yong Nie [1,2], Zhenhua Lao [3], Jiajia Liu [2,3], Yichi Huang [2,3], Xiangwei Sun [1,2,*], Jianzhong Tang [1,2] and Zheng Chen [1,2,3]

1   The State Key Laboratory of Fluid Power and Mechatronic Systems, Zhejiang University, Hangzhou 310027, China
2   Hainan Instruction, Zhejiang University, Sanya 572025, China
3   Ocean College, Zhejiang University, Zhoushan 316021, China
*   Correspondence: gfysxw@zju.edu.cn

**Abstract:** This paper presents a disturbance-observer-based sliding mode control strategy for an underwater electro-hydrostatic actuator, particularly considering that electro-hydrostatic actuators (EHAs) significantly suffer from sea pressure disturbance, which makes it hard to achieve high-precision position control. Therefore, a nonlinear disturbance observer was designed to aim at the matched and mismatched disturbance caused by sea pressure disturbance. Then, a nonlinearities model for an underwater EHA was established, and a related non-singular fast terminal sliding mode (NFTSM) controller was designed by changing the conventional sliding mode surface to further improve the control accuracy. In addition, the backstepping tool was used to guarantee the robust stability of the entire three-order hydraulic dynamic system. Finally, a comparative simulation was conducted with different load forces in AMESim and Simulink, which effectively verified the high tracking performance of the proposed control strategy.

**Keywords:** underwater elector hydrostatic actuator; disturbance observer; backstepping control; sliding-mode control

## 1. Introduction

The marine hydraulic system plays an important role in today's ocean equipment, such as underwater hydraulic manipulators [1], hydraulic steering gear, and electric hydraulic excavators [2]. However, with the demand in the improvement of exploration depth and precision, the drawbacks of traditional hydraulic systems, such as a large volume, high energy consumption and high leakage, have gradually become an obstacle for systems working perfectly in an ocean environment. Compared with traditional hydraulic systems, an electro-hydrostatic actuator (EHA) has a higher integration and lower energy consumption, which makes it a very suitable hydraulic equipment for working in the ocean filed. However, most of the current electro-hydrostatic actuators are applied in the aerospace field. If EHA is applied underwater, it must overcome two difficulties. One is the structure design adapted to the underwater environment and another is the position controller design that provides the fundamental function for working.

In the case of structure design, Liu et al. [3] added a pressure compensator to balance the pressure of the sea, and the dynamic characteristics of the pressure compensator were established. However, it only considers the matched disturbance due to the assumption of a static state for the pressure compensator. In the case of position controller design, the performance of position tracking control is unsatisfactory due to the underwater EHA system facing not only the nonlinearities of the hydraulic system but also the disturbance brought about by the underwater environment. We can classify the disturbance of seawater as matched and mismatched disturbance.

In order to deal with the matched and mismatched disturbance/uncertainty, there are several general approaches, such as sign of the error (RISE) control [4], robust control [5–8], and disturbance-observer-based control (DOBC) [9,10]. The mismatched disturbance is the main difficulty compared with matched disturbance, which has been considered by some literature. Firstly, two auxiliary error signals were introduced into the recursive backstepping design framework by Deng et al. [11–13], and the RISE feedbacks were synthesized to eliminate the matched and mismatched uncertainties simultaneously. However, the assumption conditions for disturbance are strict, and always demand that the disturbance is bounded and the second derivative exists. Robust control can suppress the mismatched disturbance using the high nonlinear gain, which is limited by the actual physical system. In addition, the disturbance observer can observe and compensate for the matched and mismatched disturbance/uncertainty. For example, Luo et al. [14] presented an extended state-observer-based (ESO) to estimate not only the unmeasured system states but also the modeling uncertainties for a hydraulic servo control system, and then designed active disturbance rejection adaptive control. Guo [15] used ESO to handle the unknown load disturbance and uncertain nonlinearity. In addition, neural network algorithms can also be used to estimate the uncertainty of the system. Seo et al. [16] used a radial basis function neural network (RBFNN) algorithm to estimate system uncertainty.

In these methods, the disturbance observer has broad application prospects in motor and hydraulic systems. However, these are always limited to the linear disturbance observer (LDO), which is unavailable for more general nonlinear systems. Thus, the nonlinear disturbance observer is of great importance for the disturbance estimation of nonlinear systems, such as [17–19]. Therefore, the nonlinear disturbance observer was used in this paper.

In order to deal with the hydraulic system high-order nonlinearities, design tools for the nonlinear system were produced, such as feedback linearization and backstepping control [20,21]. The feedback linearization and backstepping control can be used as a general design program in the process of sliding and adaptive controller design. There are many kinds of EHA control algorithms for position control, such as adaptive control [22] and sliding mode control [23,24]. Cho and Burton [22] pointed out that, when using a simple adaptive controller (SAC) in an EHA, the position tracking error is significantly reduced under the external interference load compared to traditional PID control. Wang et al. [23] applied a sliding mode controller (SMC) to a high-precision EHA position control system to study the influence of nonlinear and discontinuous friction forces on EHA position control. Chen et al. [24] researched terminal sliding mode tracking control for nonlinear systems, and obtained the terminal sliding mode control model of the SISO system. Shen et al. [25] proposed an adaptive integral terminal sliding mode controller to guarantee the robustness of the system.

Compared with the adaptive control, the sliding mode control makes the system enter the sliding mode motion state and converge to the control target quickly, which provides an effective method for the robust design of time-delay and uncertain systems. The biggest disadvantage of sliding mode control is chattering in the output of the system controller. To solve this problem, this paper used a hyperbolic tangent function instead of a sign function in exponential reaching law to reduce chattering.

The contributions of this study are summarized as follow:

1: The matched and mismatched uncertainty/disturbances of underwater EHA were observed, respectively.

2: A novel sliding mode controller with an observer was designed to overcome the unmatched and matched uncertainties.

The mathematical model of the underwater EHA is established in Section 2. The design of the disturbance observer and sliding mode controller is established in Section 3. The Simulink simulation of three controllers is established in Section 4. AMESim/Simulink simulation is established in Section 5.

## 2. Principles and Modeling

### 2.1. System Principles

The hydraulic derived principle of the underwater EHA is showed in Figure 1, which consists of a servo motor, hydraulic pump, hydraulic cylinder, pressure compensator, safety valve, flow matched valve, and pressure/position sensors. The servo motor provides power for the EHA to drive the hydraulic pump movement, and its motor speed is controlled by the integrated control unit. The hydraulic pump selected seven plunger pump because the flow pulsation of seven plunger pumps is small. The hydraulic cylinder is asymmetric and the pressure compensator is of rolling diaphragm type. The flow matched valve is connected with the tank and, when the hydraulic cylinder is working, the compensated hydraulic oil is exchanged with the tank through the flow-matched valve. The other end of the tank is connected to a pressure compensator to overcome the seawater pressure. At the same time, a pressure sensor is installed on the oil path in and out of the oil chamber, and a displacement sensor is installed on the piston rod to measure the position of piston rod.

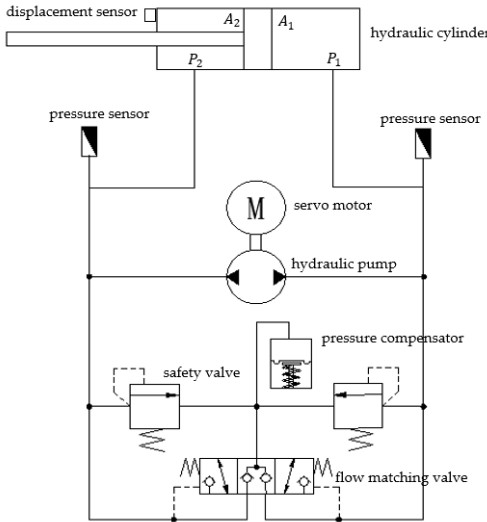

**Figure 1.** EHA hydraulic principle schematic.

### 2.2. System Modeling

Due to the closed-loop bandwidth of the servo-motor being much larger than the other parts of the system [26–28], the servo motor dynamics can be regarded as a proportional static equation:

$$\omega = ku \tag{1}$$

where $k$ is the input coefficient and $u$ is the voltage control input proportional to the pump speed $\omega$.

According to the flow continuity equation of the hydraulic pump, the variable speed pump flow rate model can be presented as

$$q_L = d_p\omega - C_{tp}p_L \tag{2}$$

where $d_p$ is the displacement of the pump, $C_{tp}$ is the total leakage coefficient of the pump, $q_L$ is the flow of the pump, and $p_L$ is the pressure of the hydraulic pump.

For the hydraulic cylinder, the pressure dynamics are established:

$$\begin{cases} Q_1 - C_{ip}(p_1 - p_2) = \frac{V_{10} + A_1 x_L}{\beta_e}\dot{p}_1 + A_1\dot{x}_L \\ -Q_2 + C_{ip}(p_1 - p_2) = \frac{V_{20} - A_2 x_L}{\beta_e}\dot{p}_2 - A_2\dot{x}_L \end{cases} \tag{3}$$

where $A_1$, $A_2$ are the effective piston areas on each side of the cylinder piston, $p_1, p_2$ are the pressure of the two chambers, respectively, $V_{10}, V_{20}$ are the initial volumes of the of the

two chambers, respectively, $x_L$ is piston displacement, $Q_1$, $Q_2$ are, respectively, the flow of the oil inlet and outlet chambers of the hydraulic cylinder, $C_{ip}$ is the leakage coefficient of the hydraulic cylinder, and $\beta_e$ is the coefficient of viscosity.

The cylinder motion dynamics can be modeled as

$$p_1 A_1 - p_2 A_2 - p_s(A_1 - A_2) = m\ddot{x}_L + B\dot{x}_L + F_t \tag{4}$$

where $p_s$ is the seawater pressure, $m$ is the total mass of the piston rod and load, $B$ is viscous damping, and $F_t$ is the external interference force.

The flow equation for seawater pressure compensation can be modeled as [29]

$$Q_c = A_0 \dot{x}_c + \frac{V_0}{\beta_e} \dot{p}_c \tag{5}$$

where $Q_c$ is the compensation quantity of the flow. The reason for this is that the effective area of the two chambers of the asymmetric hydraulic cylinder is different, so the amount of oil in and out is different ($Q_1 \neq Q_2$) when working. $A_0$ is the effective area of the rolling diaphragm and $V_0$ is the total capacity of the oil tank and pressure compensator. Since the volume change in the compensator is far less than $V_0$, $V_0 + A_0 x_c \approx V_0$.

The dynamics of seawater pressure compensation with rolling diaphragm type can be aggregately modeled as

$$p_c A_0 - p_s A_0 - k_s x_0 - k_x x_c = m_0 \ddot{x}_c + B_0 \dot{x}_c \tag{6}$$

where $k_s$ is the spring stiffness, $x_0$ is the initial displacement of the spring, $k_x$ is the total stiffness of the spring and the rolling diaphragm, $p_c$ is the pressure inside the pressure compensator, $m_0$ is the mass of the pressure compensator, and $B_0$ is the viscous damping of the pressure compensator.

### 2.3. System Formulation

The state-space variables are defined as $x_1 = x_L$, $x_2 = \dot{x}_L$, $x_3 = p_1$. When the piston extends, which is called working condition 1, the flow satisfies $Q_1 = Q_L$, $Q_2 = Q_L - Q_c$, and the pressure in the oil return chamber is equal to the pressure in the compensator $p_2 = p_c$, which are different key assumptions compared with reference [3]. The dynamics of the whole system can be modeled as

$$\dot{x}_1 = x_2 \quad \dot{x}_2 = \frac{A_1 x_3 - B x_2}{m} + F_1 \quad \dot{x}_3 = \frac{\beta_e}{V_{10} + A_1 x_1} \big[ d_p u - (C_{tp} + C_{ip}) x_3 - A_1 x_2 \big] + F_2$$

where $F_1 = -\frac{1}{m}(A_2 p_c + p_s(A_1 - A_2) + F_t)$ and $F_2 = \frac{\beta_e}{V_{10} + A_1 x_1} C_{ip} p_c$.

In contrast, defining $x_3 = p_1$ when the piston retracts, which is called working condition 2, $Q_1 = Q_L - Q_c$, $Q_2 = Q_L$, $p_1 = p_c$, and the dynamics of $x_2$ and $x_3$ are given as

$$\dot{x}_2 = \frac{-A_2 x_3 - B x_2}{m} + F_1 \quad \dot{x}_3 = \frac{\beta_e}{V_{20} - A_2 x_1} \big[ d_p \omega - (C_{tp} + C_{ip}) x_3 + A_2 x_2 \big] + F_2$$

where $F_1 = \frac{1}{m}(A_1 p_c - p_s(A_1 - A_2) - F_t)$ and $F_2 = \frac{\beta_e}{V_{20} - A_2 x_1} C_{ip} p_c$.

**Assumption 1.** $|F_1| \leq \overline{F}_1$, $|F_2| \leq \overline{F}_2$. $F_1, F_2$ *are the mismatched and matched disturbances and* $\overline{F}_1$ *and* $\overline{F}_2$ *are the upper bound of disturbances.*

The control goal of the system is to make the position of the underwater EHA accurately track the desired trajectory. Unfortunately, mismatched and matched disturbances are the main factors that influence the precision performance.

### 3. Sliding Mode Control with Disturbance Observed

In order to deal with mismatched and matched disturbances, nonlinear disturbance was designed. Then, a novel sliding mode controller with an observer was integrated. Considering that the design principles of the disturbance observer and controller are the same for the two EHA working conditions, this paper only designed the observer and controller for working condition 1 in Section 3.

*3.1. Disturbance Observer Design*

Define virtual variables $\Delta_1$ as follows [30]:

$$\Delta_1 = F_1 - \delta_1 x_2$$

where $\delta_1 > 0.5$ is a positive constant.

Define $\hat{\Delta}_1$ as the estimation of $\Delta_1$:

$$\hat{\Delta}_1 = \hat{F}_1 - \delta_1 x_2 \tag{7}$$

where $\hat{F}_1$ is the estimation of $F_1$.

$\widetilde{F}_1$ is the observation error of $F_1$ and $\widetilde{\Delta}_1$ is the observation error of $\Delta_1$.

Take the derivative of $\Delta_1$:

$$\dot{\Delta}_1 = \dot{F}_1 - \delta_1 \dot{x}_2 = \dot{F}_1 - \delta_1 \left( \frac{A_1 x_3 - B x_2}{m} + F_1 \right) = \dot{F}_1 - \delta_1 \left( \frac{A_1 x_3 - B x_2}{m} + \delta_1 x_2 \right) - \delta_1 \Delta_1 \tag{8}$$

The estimation law is given as

$$\dot{\hat{\Delta}}_1 = -\delta_1 \left[ \frac{A_1 x_3 - B x_2}{m} - \delta_1 x_2 \right] - \delta_1 \hat{\Delta}_1 \tag{9}$$

From (23) and (24), the following relationship is achieved:

$$\dot{\widetilde{\Delta}}_1 = \dot{F}_1 - \delta_1 \widetilde{\Delta}_1 \tag{10}$$

**Theorem 1.** *For systems with matched disturbances, design a virtual variable observer (10). Consider the relation (8): the perturbation observation error will eventually converge to a neighborhood near the zero.*

$$\left| \widetilde{F}_1 \right| \leq \frac{c_1}{\sqrt{2\delta_1 - 1}}$$

*where $c_1$ is a positive number close to zero.*

**Proof of Theorem 1.** The following Lyapunov function is established:

$$V = \frac{1}{2} \widetilde{F}_1^{\,2} \tag{11}$$

Taking the derivative of $V$,

$$\begin{aligned} \dot{V} &= \widetilde{F}_1 \dot{\widetilde{F}}_1 = \widetilde{\Delta}_1 \dot{\widetilde{\Delta}}_1 \\ &= \widetilde{\Delta}_1 \left[ \dot{F}_1 - \delta_1 \widetilde{\Delta}_1 \right] \\ &= -\delta \widetilde{\Delta}_1^{\,2} + \widetilde{\Delta}_1 \dot{F}_1 \end{aligned} \tag{12}$$

The inequality $\left(\widetilde{\Delta}_1 - \dot{F}_1\right)^2 = \widetilde{\Delta}_1{}^2 + \dot{F}_1{}^2 - 2\widetilde{\Delta}_1 \cdot \dot{F}_1 \geq 0$ can render into $\widetilde{\Delta}_1 \cdot \dot{F}_1 \leq \frac{1}{2}\left(\widetilde{\Delta}_1{}^2 + \dot{F}_1{}^2\right)$. Therefore, Equation (12) can be rewritten as

$$
\begin{aligned}
\dot{V} &\leq -\delta\widetilde{\Delta}_1{}^2 + \frac{1}{2}\left(\widetilde{\Delta}_1{}^2 + \dot{F}_1{}^2\right) \\
&\leq -\left(\delta - \frac{1}{2}\right)\widetilde{\Delta}_1{}^2 + \frac{1}{2}c_1{}^2 \\
&\leq -(2\delta - 1)\frac{1}{2}\widetilde{\Delta}_1{}^2 + \frac{1}{2}c_1{}^2 \\
&\leq -(2\delta - 1)\frac{1}{2}\widetilde{F}_1{}^2 + \frac{1}{2}c_1{}^2 \\
&\leq -(2\delta - 1)V + \frac{1}{2}c_1{}^2
\end{aligned}
\tag{13}
$$

The convergence of the observation error $\widetilde{F}_1$ requires condition $\dot{V} \leq -(2\delta - 1)V + \frac{1}{2}c_1{}^2 \leq 0$. Integrate both sides of inequality (13):

$$
\ln \frac{-(2\delta - 1)V + \frac{c_m^2}{2}}{-(2\delta - 1)V_0 + \frac{c_m^2}{2}} \leq -(2\delta - 1)t
\tag{14}
$$

where $V_0$ is the initial value of $V$.

Inequality (14) can be simplified as

$$
V \leq V_0 e^{-(2\delta - 1)t} + \frac{c_m^2}{2(2\delta - 1)}
\tag{15}
$$

Combined with Equation (11), Inequality (15) can be transformed into

$$
\left|\widetilde{F}_1\right| \leq \sqrt{2V_0 e^{-(2\delta - 1)t} + \frac{c_m^2}{2\delta - 1}}
\tag{16}
$$

As time $t$ increases, $e^{-(2\delta - 1)t}$ will gradually tend to zero. Therefore, $\left|\widetilde{F}_1\right| \leq \frac{c_m}{\sqrt{2\delta - 1}}$ is satisfied.

The estimation for $F_2$ is as follows:

$$
\hat{F}_2 = \hat{\Delta}_2 + \delta_2 x_3 \dot{\hat{\Delta}}_2 = -\delta_2\left[\frac{\beta_e}{V_{10} + A_1 x_1}\left[d_p u - \left(C_{tp} + C_{ip}\right)x_3 - A_1 x_2\right] - \delta_2 x_3\right] - \delta_2\hat{\Delta}_2
\tag{17}
$$

□

**Theorem 2.** *For systems with mismatched disturbances, design a virtual variable observer (17). The perturbation observation error will eventually converge to a neighborhood near the zero.*

$$
\left|\widetilde{F}_2\right| \leq \frac{c_2}{\sqrt{2\delta_2 - 1}}.
$$

*where* $c_2$ *is a positive number close to zero.*

**Proof of Theorem 2.** The proof process of Theorem 2 is the same as Theorem 1. □

*3.2. Backstepping Sliding Mode Control (BSMC)*

Suppose that the position command is $x_d$.

Step 1:

Define $z_1 = x_1 - x_d$, $\dot{z}_1 = x_2 - \dot{x}_d$, $z_2 = x_2 - \dot{x}_d + c_1 z_1$.

The following Lyapunov function is established:

$$
V_1 = \frac{1}{2}z_1^2
\tag{18}
$$

$$\dot{V}_1 = z_1 \dot{z}_1 = z_1 (x_2 - \dot{x}_d) = -c_1 z_1^2 + z_1 z_2 \tag{19}$$

$z_2 = 0$, $V_1$ can be positive definite and $\dot{V}_1 < 0$.

Step 2:

The following Lyapunov function is established:

$$V_2 = V_1 + \frac{1}{2} z_2^2 \tag{20}$$

$$\begin{aligned}
\dot{V}_2 &= -c_1 z_1^2 + z_1 z_2 + z_2 (\dot{x}_2 - \ddot{x}_d + c_1 \dot{z}_1) \\
&= -c_1 z_1^2 + z_1 z_2 + z_2 \left( \frac{1}{m}(A_1 x_3 - Bx_2 - mF_1) - \ddot{x}_d + c_1 \dot{z}_1 \right)
\end{aligned} \tag{21}$$

Define $x_3 = \frac{1}{A_1} \left( Bx_2 + mF_1 - mc_1 \dot{z}_1 + m\ddot{x}_d - mz_1 - mc_2 z_2 + mz_3 \right)$, $c_2 > 0$.

$$z_3 = \frac{A_1}{m} x_3 - \frac{B}{m} x_2 + F_1 + c_1 \dot{z}_1 + z_1 + c_2 z_2 - \ddot{x}_d \tag{22}$$

$$\dot{V}_2 = -c_1 z_1^2 - c_2 z_2^2 + z_2 z_3 \tag{23}$$

Step 3:

The sliding mode surface function is defined as

$$s = k_1 z_1 + k_2 z_2 + z_3 \tag{24}$$

The following Lyapunov function is established:

$$V_3 = V_2 + \frac{1}{2} s^2 \tag{25}$$

$$\begin{aligned}
\dot{V}_3 &= -c_1 z_1^2 - c_2 z_2^2 + z_2 z_3 + s(k_1 \dot{z}_1 + k_2 \dot{z}_2 + \dot{z}_3) \\
&= -c_1 z_1^2 - c_2 z_2^2 + z_2 z_3 + s \Big[ k_1 \dot{z}_1 + k_2 \dot{z}_2 \\
&\quad + \frac{A_1}{m} \left( \frac{\beta_e}{V_{10} + A_1 x_1} \left( d_p u - (C_{tp} + C_{ip}) x_3 - A_1 x_2 \right) + \hat{F}_2 \right) \\
&\quad - \frac{B}{m} \left( \frac{A_1 x_3 - Bx_2}{m} + \hat{F}_1 \right) + \dot{\hat{F}}_1 + c_1 \ddot{z}_1 + \dot{z}_1 + c_2 \dot{z}_2 - \dddot{x}_d \Big]
\end{aligned} \tag{26}$$

The backstepping sliding mode controller can be designed:

$$\begin{aligned}
u &= \frac{1}{d_p} \Big[ A_1 x_2 + (C_{tp} + C_{ip}) x_3 + \frac{V_{10} + A_1 x_1}{\beta_e} \Big( -\hat{F}_2 + \frac{m}{A_1} \Big( \frac{B}{m} \Big( \frac{A_1 x_3 - Bx_2}{m} + \hat{F}_1 \Big) \\
&\quad -k_1 \dot{z}_1 - k_2 \dot{z}_2 - \dot{\hat{F}}_1 - c_1 \ddot{z}_1 - \dot{z}_1 - c_2 \dot{z}_2 + \dddot{x}_d - h(s + \beta \operatorname{sign}(s)) \Big) \Big) \Big]
\end{aligned} \tag{27}$$

Substitute (27) into (26):

$$\dot{V}_3 = -c_1 z_1^2 - c_2 z_2^2 + z_2 z_3 - hs^2 - h\beta |s| < 0 \tag{28}$$

**Proof of Stability.** Define the matrix $Q$ [31]

$$Q = \begin{bmatrix} hk_1^2 + c_1 & hk_1 k_2 & hk_1 \\ hk_1 k_2 & hk_2^2 + c_2 & hk_2 - \frac{1}{2} \\ hk_1 & hk_2 - \frac{1}{2} & h \end{bmatrix} \tag{29}$$

$$\begin{aligned}
Z^T Q Z &= \begin{bmatrix} z_1 & z_2 & z_3 \end{bmatrix} \begin{bmatrix} hk_1^2 + c_1 & hk_1 k_2 & hk_1 \\ hk_1 k_2 & hk_2^2 + c_2 & hk_2 - \frac{1}{2} \\ hk_1 & hk_2 - \frac{1}{2} & h \end{bmatrix} \begin{bmatrix} z_1 \\ z_2 \\ z_3 \end{bmatrix} \\
&= c_1 z_1^2 + c_2 z_2^2 - z_2 z_3 + hs^2
\end{aligned} \tag{30}$$

When $Q$ is positive definite, (30) can translate into

$$\dot{V}_3 \leq -Z^T Q Z - h\beta|s| \leq 0 \tag{31}$$

$$|Q| = c_1 h(k_2 + c_2) - \frac{c_1 + hk_1^2}{4} \tag{32}$$

Therefore, taking the appropriate $c_1, c_2, k_1, k_2, h$ can make $|Q| > 0$ and ensures that $\dot{V}_3 < 0$.

According to the LaSalle invariance principle, when $\dot{V}_3 \equiv 0$, $z$ and $s$ are identically equal to 0. Therefore, when $t \to \infty$, $z \to 0$, and $s \to 0$, $x_1 \to x_d$. $\square$

*3.3. Backstepping Nonsingular Fast Terminal Sliding Mode Control Based on Hyperbolic Tangent Function (BNFTSM)*

Because of the linear sliding surface of BSMC, the higher-order EHA systems cannot converge in finite time. Terminal sliding mode control can make higher-order systems converge in finite time, but it needs to solve the singularity problem. Thus, a backstepping nonsingular fast terminal sliding mode is used. The linear sliding mode surface is replaced by a non-singular fast terminal sliding mode surface [32,33], and the hyperbolic tangent function $tanh(s/0.05)$ is used to replace the sign function $sign(s)$ in the exponential reaching law to reduce chattering.

The sliding mode surface function is defined as

$$s = \int_0^t z_3 dt + \alpha_1 \left| \int_0^t z_3 dt \right|^{k_1} sign\left( \int_0^t z_3 dt \right) + \alpha_2 |z_3|^{k_2} sign(z_3) \tag{33}$$

In the BNFTSM sliding stage, when the system error variable is far away from the equilibrium point, the higher-order term of $\left| \int_0^t z_3 dt \right|$ plays a major role. Otherwise, the higher-order term of $z_3$ plays a major role. The combination of the two can make the system error variable converge quickly to the equilibrium point along the sliding surface (S = 0) in a finite time $t_s$ [32]. The proof of finite time $t_s$ is presented in [32].

Substitute (33) into (25):

$$\begin{aligned}
\dot{V}_3 &= -c_1 z_1^2 - c_2 z_2^2 + z_2 z_3 + s\dot{s} \\
&= -c_1 z_1^2 - c_2 z_2^2 + z_2 z_3 + s\left[ z_3 + \alpha_1 k_1 \left| \int_0^t z_3 dt \right|^{k_1-1} z_3 \right. \\
&\left. + \alpha_2 k_2 |z_3|^{k_2-1} \left( \frac{A_1}{m} \left( \frac{\beta_e}{V_{10}+A_1 x_1} \left( d_p u - (C_{tp} + C_{ip}) x_3 - A_1 x_2 \right) + \hat{F}_2 \right) \right.\right. \\
&\left.\left. - \frac{B}{m} \left( \frac{A_1 x_3 - B x_2}{m} + \hat{F}_1 \right) + \dot{\hat{F}}_1 + c_1 \ddot{z}_1 + \dot{z}_1 + c_2 \dot{z}_2 - \dddot{x}_d \right) \right]
\end{aligned} \tag{34}$$

The backstepping nonsingular fast terminal sliding mode controller can be designed:

$$\begin{aligned}
u = \frac{1}{d_p} \left[ A_1 x_2 + (C_{tp} + C_{ip}) x_3 + \frac{V_{10}+A_1 x_1}{\beta_e} \right. \\
\left( -\hat{F}_2 + \frac{m}{A_1} \left( \frac{B}{m} \left( \frac{A_1 x_3 - B x_2}{m} + \hat{F}_1 \right) - k_1 \dot{z}_1 - k_2 \dot{z}_2 - \dot{\hat{F}}_1 - c_1 \ddot{z}_1 \right. \right. \\
\left. \left. \left. -\dot{z}_1 - c_2 \dot{z}_2 + \dddot{x}_d - \frac{1}{\alpha_2 k_2} \left( |z_3|^{2-k_2} + \alpha_1 k_1 \left| \int_0^t z_3 dt \right|^{k_1-1} |z_3|^{2-k_2} + h\left( s + \beta tanh\left( \frac{s}{0.05} \right) \right) \right) \right) \right) \right]
\end{aligned} \tag{35}$$

Substitute (35) into (34):

$$\dot{V}_3 = -c_1 z_1^2 - c_2 z_2^2 + z_2 z_3 - hs^2 - h\beta stanh\left( \frac{s}{0.05} \right) < 0 \tag{36}$$

**Lemma 1.** *For$\forall x \in R, \varepsilon > 0$, the following inequality is true:*

$$xtanh\left( \frac{x}{\varepsilon} \right) = \left| xtanh\left( \frac{x}{\varepsilon} \right) \right| = |x||tanh\left( \frac{x}{\varepsilon} \right)| \geqslant 0 \tag{37}$$

*The explanation is as follows:*

$$x\tanh\left(\frac{x}{\varepsilon}\right) = x\frac{\mathrm{e}^{\frac{x}{\varepsilon}} - \mathrm{e}^{-\frac{x}{\varepsilon}}}{\mathrm{e}^{\frac{x}{\varepsilon}} + \mathrm{e}^{-\frac{x}{\varepsilon}}} = \frac{1}{\mathrm{e}^{2\frac{x}{\varepsilon}} + 1} x\left(\mathrm{e}^{2\frac{x}{\varepsilon}} - 1\right) \tag{38}$$

*This is due to the following inequality:*

$$\begin{cases} \mathrm{e}^{\frac{2x}{\varepsilon}} - 1 \geqslant 0, & x \geqslant 0 \\ \mathrm{e}^{\frac{2x}{\varepsilon}} - 1 < 0, & x < 0 \end{cases} x\left(\mathrm{e}^{\frac{2x}{\varepsilon}} - 1\right) \geqslant 0 \tag{39}$$

*Equation (40) can be translated as*

$$x\tanh\left(\frac{x}{\varepsilon}\right) = \frac{1}{e^{2\frac{x}{\varepsilon}} + 1} x\left(e^{2\frac{x}{\varepsilon}} - 1\right) \geqslant 0 \tag{40}$$

**Proof of Stability.** Define the matrix $Q$:

$$Q = \begin{bmatrix} hk_1^2 + c_1 & hk_1k_2 & hk_1 \\ hk_1k_2 & hk_2^2 + c_2 & hk_2 - \frac{1}{2} \\ hk_1 & hk_2 - \frac{1}{2} & h \end{bmatrix} \tag{41}$$

$$Z^TQZ = \begin{bmatrix} z_1 & z_2 & z_3 \end{bmatrix} \begin{bmatrix} hk_1^2 + c_1 & hk_1k_2 & hk_1 \\ hk_1k_2 & hk_2^2 + c_2 & hk_2 - \frac{1}{2} \\ hk_1 & hk_2 - \frac{1}{2} & h \end{bmatrix} \begin{bmatrix} z_1 \\ z_2 \\ z_3 \end{bmatrix} = c_1z_1^2 + c_2z_2^2 - z_2z_3 + hs^2 \tag{42}$$

When Q is positive definite, (30) can translate into

$$\dot{V}_3 \leq -Z^TQZ - h\beta tanh\left(\frac{s}{0.05}\right) \leq 0 \tag{43}$$

$$|Q| = c_1h(k_2 + c_2) - \frac{c_1 + hk_1^2}{4} \tag{44}$$

Therefore, taking the appropriate $c_1, c_2, k_1, k_2, h$ can make $|Q| > 0$ and ensures that $\dot{V}_3 < 0$.

According to the LaSalle invariance principle, when $\dot{V}_3 \equiv 0$, $z$ and $s$ are identically equal to 0. Therefore, when $t \to t_s$, $z \to 0$, and $s \to 0$, $x_1 \to x_d$. $\square$

## 4. Simulation Results

### 4.1. Configuration of Simulations

The performance of the proposed method was first evaluated using Simulink, in which, the hydraulic system model was established with the S-function-based equation of state space in Section 2.3. In addition, the control algorithm was also programmed with S-function-based Equations (27) and (35). Furthermore, the AMESim model of the hydraulic system was established in Simcenter AMESim, shown in Figure 2, which allows the designer to integrate hydraulic system models with a control system in order to assess the actuator performance at large.

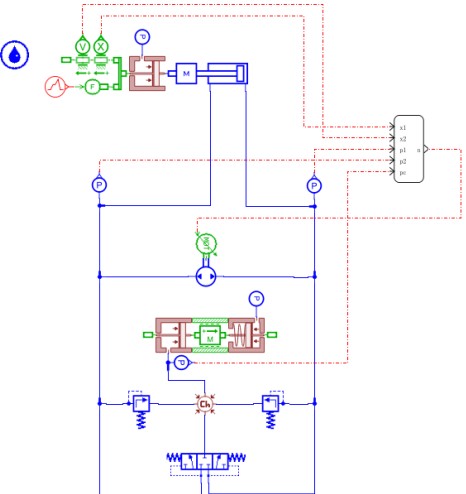

**Figure 2.** AMESim model.

The three controllers, PID, BSMC, and BNFTSM, were, respectively, simulated under the conditions of 5 MPa water pressure and different load forces. The control gains were adjustedto obtain the best tracking performance in both methods. The controller parameters were designed as

PID: The control parameters were obtained by pole zero assignment. $k_p$ is 10,000, $k_i$ is 200, and $k_d$ is 0.

BSMC: According to Equation (32), $|Q|$ had to be greater than 0. In addition, the control parameters could not be too large, which could make the controller used in practice. The values of $k_1$, $k_2$, $c_1$, and $c_2$ were usually set between 0 and 100. Thus, $k_1$ is 10, $k_2$ is 10, $c_1$ is 60, $c_2$ is 60, $h$ is 10, and $\beta$ is 1.5.

BNFTSM: The first part was the same as with BSMC, and $k_1$ and $k_2$ must satisfy the following inequality:

$$\begin{cases} 1 < k_2 < 2 \\ \quad k_1 > 1 \end{cases}$$

Thus, $\alpha_1$ is 10, $\alpha_2$ is 5, $k_1$ is 1.8, $k_2$ is 1.2, $c_1$ is 200, $c_2$ is 300, $h$ is 10, and $\beta$ is 1.5.
The loads forces were designed as

$$\text{Case1 } Ft = 4000 \text{N} \quad \text{Case2 } Ft = 4000 \sin(t) \text{N}$$

The hydraulic system model parameters used in the Simulink and AMESim are listed in Table 1.

**Table 1.** Model parameters used in the Simulink.

| Parameter | Value | Parameter | Value |
|:---:|:---:|:---:|:---:|
| $m$ | 100 kg | $m_0$ | 1 kg |
| $B$ | 2000 N/(m/s) | $B_0$ | 1270 N/(m/s) |
| $V_{10}$ | $1.852 \times 10^{-4}$ m$^3$ | $k_x$ | 3100 N/m |
| $V_{20}$ | $5.937 \times 10^{-4}$ m$^3$ | $C_{tp}$ | $3 \times 10^{-11}$ |
| $d_p$ | $1 \times 10^{-6}$ | $C_{ip}$ | $5 \times 10^{-11}$ |
| $\beta_e$ | $7.5 \times 10^8$ | $p_s$ | $5 \times 10^6$ Pa |

### 4.2. Comparative Analysis

The tracking performance of the controllers are shown in Figures 3–5 based on the Simulink model. The control effect of BNFTSM is better than PID and BSMC under different load forces, and can suppress the sudden increase error compared with BSMC. When the load is 4000N and $4000 \sin(t)$N, the maximum tracking error for BNFTSM can be controlled

at $3 \times 10^{-4}$ m; that is, the control accuracy is less than or equal to 0.12%. The nonlinear observer can accurately compensate for the mismatched and matched disturbance as shown in Figure 6, which depicts the observer errors for those disturbances. The observer accuracy for mismatched disturbance is approximately 0.18FS%, in which, the full scale is $1 \times 10^{-2}$. The observer accuracy for matched disturbance is 0.2%Fs, in which, the full scale is $1 \times 10^{10}$.

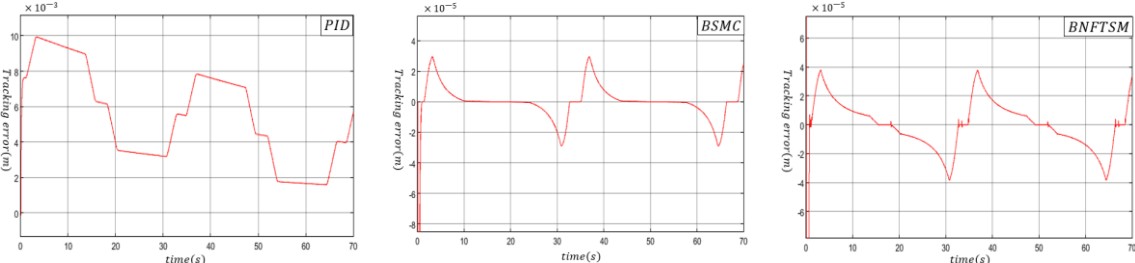

**Figure 3.** Tracking error curves of three controllers without load force.

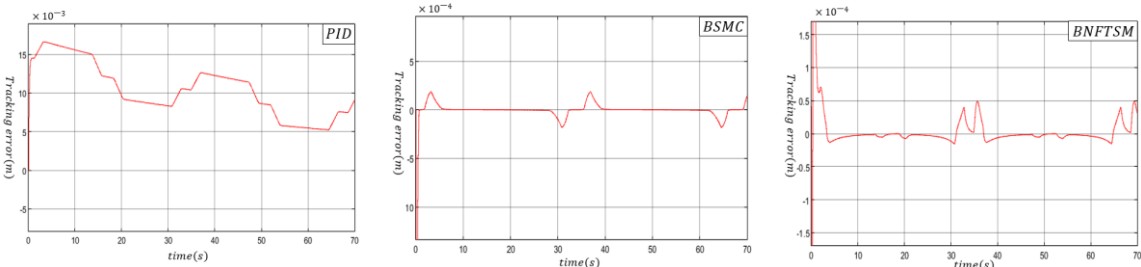

**Figure 4.** Tracking error curves of three controllers under 4000N constant load force.

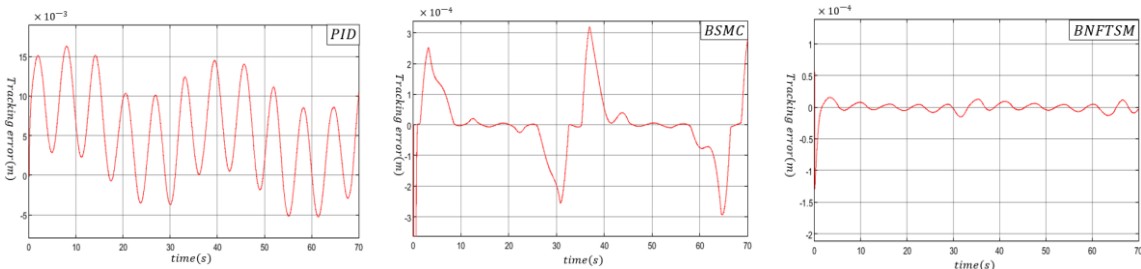

**Figure 5.** Tracking error curves of three controllers under variable load force of $4000 \sin(t)$N.

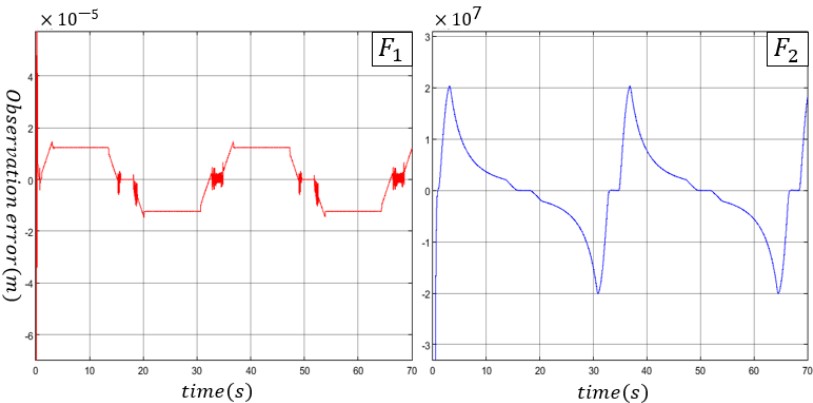

**Figure 6.** Observation error of $F_1$ and $F_2$.

The tracking-performance-based AMESim model is shown in Figures 7–9. Because more factors are considered in the design of AMESim model parameters, such as the dead

zone of the hydraulic cylinder and the dynamic characteristics of the valve, it is more accurate than the Simulink model. Therefore, the control effect of the controller will be different. The control effect of BNFTSM and BSMC is better than PID under case 1 and case 2. Under a constant load, the control accuracy of BNFTSM ($2 \times 10^{-5}$ m) is better than that of BSMC ($1 \times 10^{-4}$ m), but BSMC control is more stable and BNFTSM oscillates more violently. BNFTSM performs better than BSMC under a time-varying load.

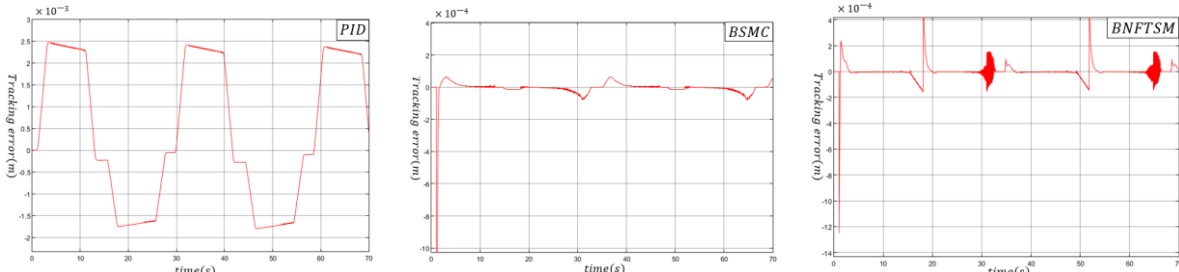

**Figure 7.** Tracking error curves of three controllers without load force: PID, BSMC, BNTFSM.

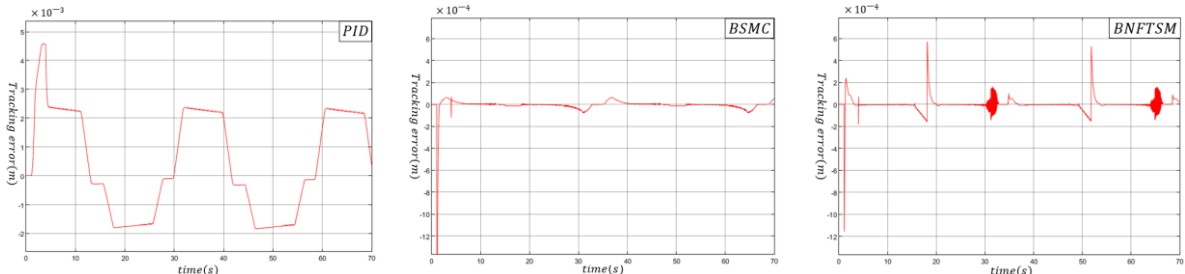

**Figure 8.** Tracking error curves of three controllers under 4000N constant load force: PID, BSMC, BNTFSM.

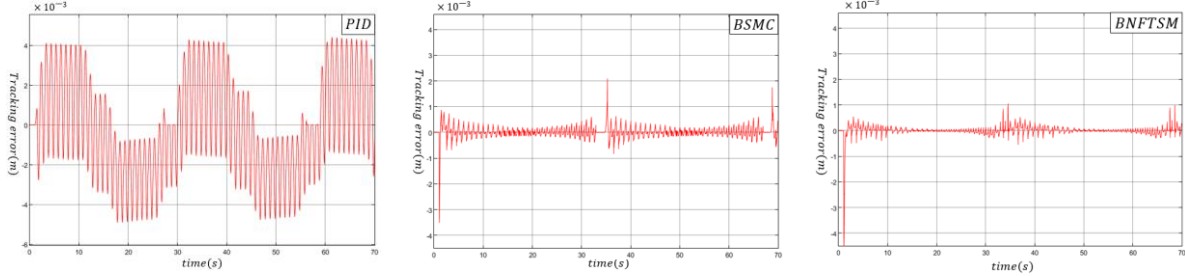

**Figure 9.** Tracking error curves of three controllers under variable load of $4000\sin(t)$N: PID, BSMC, BNTFSM.

## 5. Conclusions

Aiming at the high precision control of an underwater electro-hydrostatic actuator, a non-singular terminal sliding mode controller based on a disturbance observer was designed. It is worth mentioning that the matched and unmatched disturbances were overcome. In addition, by changing the equation of the sliding mode surface, its control accuracy was further improved, which was verified by the simulation of AMESim and Simulink under time-varying load conditions. The control strategy plays an important role in the underwater EHA system, which can work perfectly in the ocean exploration field. In further studies, synchronized control for multiple actuators [34,35] and fault-tolerant control [36] will be conducted.

**Author Contributions:** Conceptualization, Z.C. and Y.N.; methodology, Z.L. and Y.H.; software, X.S.; validation, Y.N., J.T., Z.L. and J.L.; formal analysis, Z.L.; investigation, X.S.; resources, Y.N.; data curation, Z.L.; writing—original draft preparation, J.L.; writing—review and editing, J.L. and Z.C.; project administration, Y.N.; funding acquisition, Y.N. All authors have read and agreed to the published version of the manuscript.

**Funding:** This work is supported by the Hainan Special PhD Scientific Research Foundation of Sanya Yazhou Bay Science and Technology City (No. HSPHDSRF-2022-04-004), Hainan Provincial National Natural Science Foundation of China (No. 521MS065), and supported by Scientific Research Fund of Zhejiang University (No. XY2022020).

**Data Availability Statement:** Not applicable.

**Conflicts of Interest:** The authors declare no conflict of interest.

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
