# Peer review of "Disturbance Observer-Based Sliding Mode Controller for Underwater Electro-Hydrostatic Actuator Affected by Seawater Pressure"

_machines, doi:10.3390/machines10121115_

Round 1

Reviewer 1 Report

The paper is too much theoretical. The control methods are presented with Theorems and related Proofs, but the aim of the papers should be to explain how to apply the control methods in real application as the control of an actuator and not to explain theory.

In addition, I would like to add the following item not clear or wrong:

Line 132: Q1 and Q2 are not pressure but flow

Line 154: it is not clear the choice of the state variable. Why not only xL and xdotL but also p1. It should be useful an explanation

Line 177: it is not clear what is the working condition 1

Line 312-314: how the control parameters have been designed? The focus on the paper should be on this item!!

Line 323-337: the difference of control tracking performances between plant model made with Simulink and plant model made with AMESIM must be explained. Why they are different? Is this because of one model is more accurate then the other? Is the instability due to the model which has calculation solving problem or due to the control strategy not well tuned?

Author Response

We would first like to thank the reviewers for providing comments that have helped us to improve the comprehension of our manuscript and to add some comparison extensive of the controller with other ones known from the literature. We agree with the Associate Editor and the reviewers that more explanations w.r.t. the indicated aspects are necessary. We next give a brief summary of the performed changes, which are marked up using the “Track Changes” function. As proposed by the reviewers, we have performed the following changes to the original manuscript.

Reviewer 2 Report

This paper introduced a new method for pressure compensation for stable operation of underwater EHA. The method is well described with sufficient mathematical details. However, the discussion should be refined to ensure all major conclusions are supported by the derivatives and simulation results. Simulation results are claimed to verify the high tracking performance of the proposed control strategy. However, this conclusion is not explicitly supported by the simulation results. Specifically, authors need to address concerns below

1. Comparing BSMC and BNFTSM, the advantageous of BNFTSM is not clearly shown in math. The authors pointed out "the control effect of BMSC is not good enough... on p7", but should give more proof for this. For example, by comparing Eq.28 and Eq.36.  

2. The full name of EHA should be given in the abstract. 

3. Descriptions of the inset in Fig. 7-9 should be given in the captions. Units in the insets are not discernible. 

4. The manuscript needs extensive gramma and format checking. 

Round 2

Reviewer 1 Report

The revised version is now acceptable